# 3D U-Net Based Semantic Segmentation of Kidneys and Renal Masses on Contrast-Enahanced CT

Mingyang Zang[1], Artur Wysoczanski[1], Elsa Angelini[1,2], and Andrew F. Laine[1]

[1] Department of Biomedical Engineering, Columbia University, New York, NY, USA
[2] ITMAT Data Science Group, NIHR Imperial BRC, Imperial College, London, UK
mz2846@columbia.edu

**Abstract.** The accurate, automated detection and segmentation of renal tumors is of great interest for the imaging-based diagnosis, histologic subtyping, and management of suspected renal malignancy. The KiTS21 Grand Challenge provides 300 contrast enhanced CT images with kidney, tumors and cysts with corresponding manual annotation, to facilitate the development of robust segmentation algorithms for this task. In this work, we present an adaptation of the historically-successful 3D U-Net architecture, combined with deep supervision, foreground oversampling and large-scale image context, and trained on the majority-prediction segmentation masks. We achieve validation performance of 96.3%, 85.6%, and 83.5% volumetric Dice score, and 91.9%, 74.9% and 72.9% surface Dice score, on combined foreground, renal masses, and renal tumors, respectively.

**Keywords:** 3D U-Net · Medical Image Segmentation.

## 1  Introduction

With the increasing quantity and quality of volumetric medical images, deep-learning methods are gaining in popularity, and equal or exceed the performance of expert human reviewers on a wide variety of detection, segmentation and classification tasks [8]. Automated tumor detection and delineation is of particular interest in the context of renal masses, which are often incidentally detected and whose imaging features have significant implications for patient management [6].

3-D encoder-decoder networks, such as 3D U-Net [9] and V-Net [5], have been shown to be robust to a wide variety of segmentation tasks across imaging modalities and protocols. Crucially for medical imaging applications, where the amount of training data is often severely limited due to privacy concerns and time- and cost-prohibitive annotation, such networks can generally be trained end-to-end from very few images. Although the U-Net architecture has subsequently been augmented by the incorporation of residual blocks, attention gating, and other features, the "vanilla" U-Net often outperforms its successors.

Motivated by this observation, nnU-Net (short for "no new U-Net") [4] focuses primarily on standard network architectures, while tuning hyperparameters such as batch size, optimizer parameters, and patch and kernel size to improve the network generalization ability. The nnU-Net has achieved top performance by mean Dice similarity on all but one class of the Medical Segmentation Decathlon challenge[1], and was also the basis for the top-performing entry in the KiTS19 Grand Challenge [2] [3]. Medical images are mostly single channeled and less diverse than natural images [7]. Hence, we hypothesize that a straightforward 3D U-Net architecture with proper processing and sampling of original data and minor modifications to network architecture, may achieve similarly high performance on the current challenge dataset.

## 2    Methods

### 2.1    Training and Validation Data

Our submission made use of the official KiTS21 training set alone. The data was trained with 240 cases and validated using the remaining 60 cases. Network training and validation are performed on the majority-voting segmentation masks provided by the challenge organizers.

### 2.2    Preprocessing

The original CT scans are resampled to isotropic $1.99mm \times 1.99mm \times 1.99mm$ resolution by third-order spline interpolation, and the ground-truth segmentation masks are resampled using nearest-neighbor interpolation.

We achieved optimal network performance by performing case-by-case clipping of Hounsfield intensity values to the 0.5th and 99.5th percentile, after which each volume was normalized by subtracting the mean intensity and dividing by the standard deviation. All results reported in this paper were obtained with this normalization technique. Data augmentation including mirroring, rotation about all axes, brightness control, gamma correction, contrast adjustment, and scaling were applied randomly at run time to all image patches, with a probability of 0.4 for each operation.

### 2.3    Network architecture

The network is constructed based on the 3D U-Net [9] framework, using the nn-UNet framework; the network architecture is depicted in Fig. 1. The breadth of first convolutional block is set to 32 channels, and doubles at each downsampling step. Downsampling is continued until reaching output dimension 4x4x4. At the shallowest layers of the upsampling arm, we generate downsampled prediction masks by convolution followed by softmax output. Such supervision enables the network to predict correctly starting from the low resolution and avoid passing wrong segmentation information to the higher resolution layers.

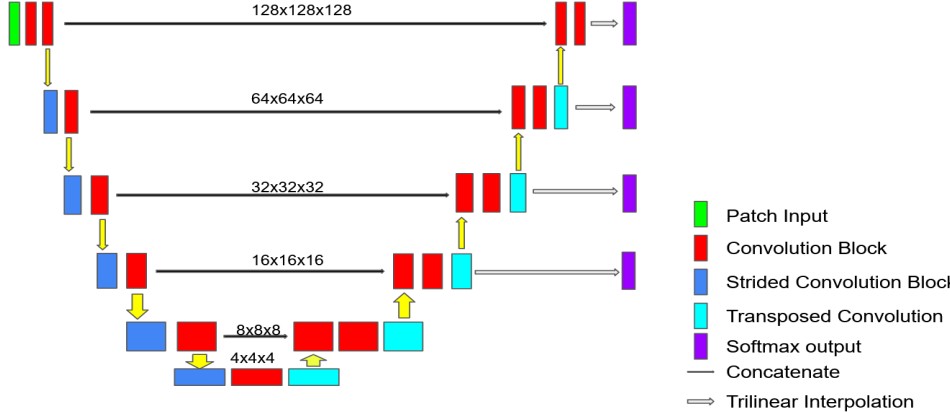

Fig. 1: The scheme of the 3D U-Net. All convolution blocks use 3x3x3 kernels and the transposed convolution use 2x2x2 kernels.

## 2.4  Loss function

We use the sum of categorical cross-entropy and soft Dice loss as our objective function at each output layer. The final loss function is the weighted sum of the losses calculated at each output layer of the network, with the weight decreasing by a factor of two with each drop in resolution.

## 2.5  Optimization strategy

We choose a patch size of $128 \times 128 \times 128$ and set the batch size to 4, maximizing the patch volume under the constraints imposed by GPU memory. Patches are sampled at random from the training images at run time, with oversampling of the foreground classes achieved by requiring that at least one-third of each patch be occupied by a foreground label (kidney, cyst, or tumor) to focus network training on the foreground classes. We implement SGD with an initial learning rate of 0.06, 0.99 momentum and Nesterov as our optimizer. We define one epoch as 250 batches and the whole training phase lasts for 1000 epochs.

## 2.6  Validation

Our hold-out validation set consists of 60 cases selected at random without replacement from the public dataset. Validation is performed using a sliding window approach with a stride equal to half the patch size.

## 2.7   Post-processing

The predictions are resampled to their original resolutions by nearest-neighbor interpolation without any further processing.

# 3   Results

Validation loss was minimized at epoch 892; training and validation loss curves are depicted in Figure 2 below.

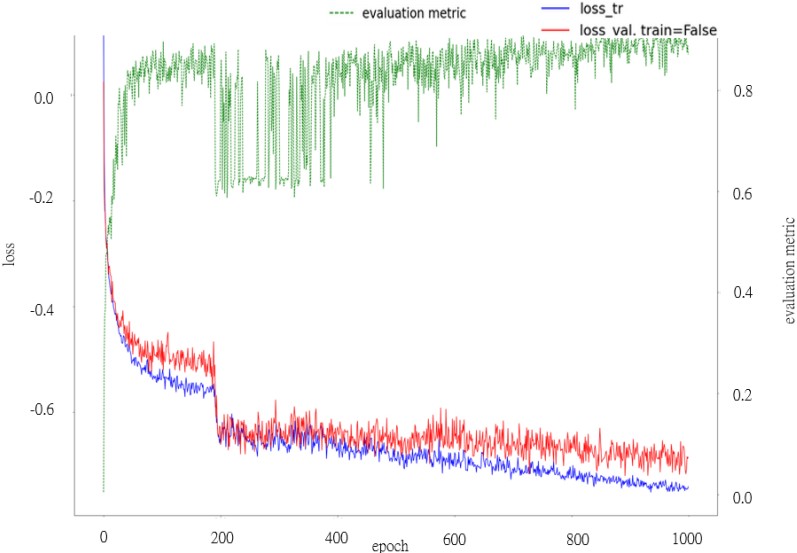

Fig. 2: Training loss (blue), validation loss (red), and exponential moving average of the composite Dice (green) across 1000 epochs.

The optimized network attained a mean volumetric Dice score of 0.963, 0.856 and 0.835 on the "kidney + masses", "masses" and "tumor" classes, respectively; mean surface Dice scores on the same classes reached 0.919, 0.749 and 0.729 respectively. Validation performance metrics are shown in Table 1 below, with representative slices from the best-performing validation cases presented in Figure 3.

Table 1: Volume Dice and Surface Dice of the validation set.

|                 | Dice(Majority/Individual) | Surface Dice(Majority/Individual) |
|-----------------|---------------------------|-----------------------------------|
| Kidney + Masses | 0.971/0.963               | 0.937/0.919                       |
| Masses          | 0.861/0.856               | 0.762/0.749                       |
| Tumor           | 0.840/0.835               | 0.743/0.729                       |

Fig. 3: Representative segmentation from the top-four validation cases, as determined by the mean of volumetric and surface Dice scores across all evaluation classes. Kidneys are annotated in red, renal tumors in green, and cysts in blue. Top row: predicted segmentation; Bottom row: majority-vote ground truth.

Both the mean Dice coefficient and mean surface Dice are relatively robust to the choice of majority or individual annotation for evaluation, with a maximum absolute decrease of 0.008 for Dice coefficient and 0.018 for surface Dice across all hierarchical classes when comparing individual annotation to the majority-voting scheme. Validation performance under the two evaluation methods was also strongly correlated at the subject level (Spearman r = 0.996).

The distributions of volumetric and surface Dice coefficients across the validation set (Figure 4) are left-skewed with multiple outliers at low Dice coefficient, most prominently among masses and tumors. Among validation cases containing a solitary renal tumor, we observed that all outliers with respect to volumetric and surface Dice have tumor volumes below the median $(24.75 \text{ cm}^3)$, as computed from the majority-vote segmentation (Figure 5). Performance on tumors below the median volume differ from performance on larger tumors, both by volumetric Dice (median 0.832, vs. 0.942, Mann-Whitney U=28, $p < 10^{-9}$ ) and surface Dice coefficients (median 0.751 vs. 0.829, Mann-Whitney U=216, $p < 0.005$).

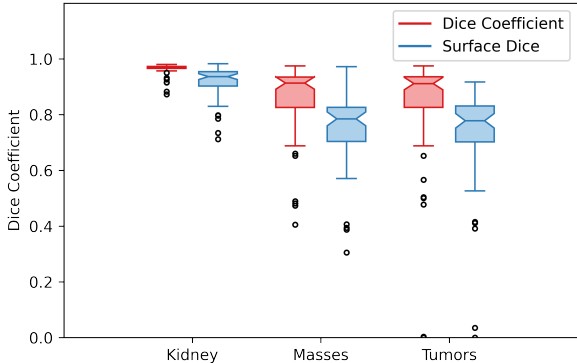

Fig. 4: Box-and-whisker plots for the volumetric and surface Dice coefficients of validation cases, for all evaluation classes.

Among cases in the validation set with worst tumor Dice coefficient, we observe three general failure modes for both tumor and cyst segmentation (Figure 6): omission or false detection of small masses (Figure 6A,D), under- segmentation (Figure 6B,C) and tumor/cyst mis-classification (Figure 6D,E). Generally, small cysts were particularly vulnerable to under-segmentation or omission.

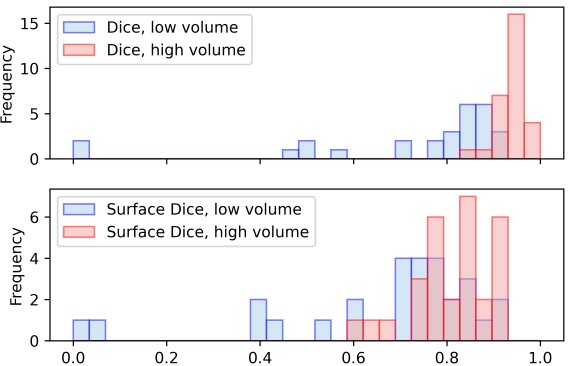

Fig. 5: Histograms of tumor volumetric (top) and surface (bottom) Dice coefficients, in validation cases with a single renal tumor (n=57). For illustrative purposes, we define the "low volume" class to include tumors with majority-vote volume below the median on this dataset, with "high volume" containing the remainder.

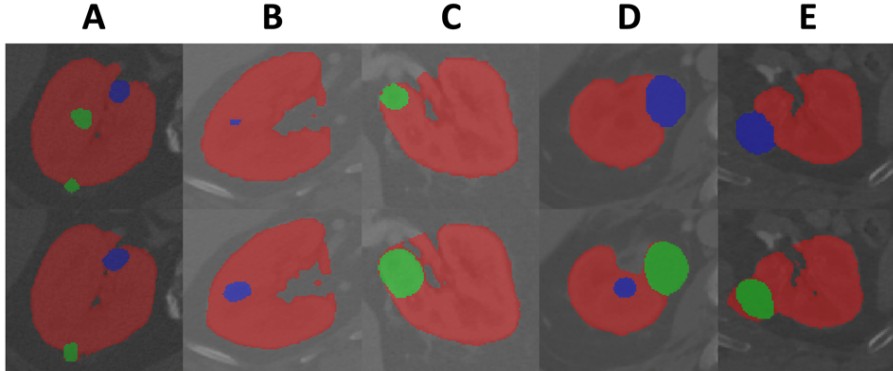

Fig. 6: Representative examples of segmentation failure, defined by lowest volumetric Dice score for the tumor class. Kidneys are annotated in red, renal tumors in green, and cysts in blue. Top row: predicted mask; Bottom row: majority-vote ground truth segmentation. Note missed/false-positive small masses (cases A,D), under-segmented cysts and tumors (cases B,C), and misclassified masses (cases D,E).

## 4    Discussion and Conclusion

Our work demonstrates once again that 3D U-Net architectures achieve competitive performance on kidney and renal mass segmentation in the KiTS21 dataset. Building on the nnU-Net framework, we incorporate deep supervision, a targeted foreground-oversampling strategy, and large-volume image patches with maximized batch size to optimize network performance. On the public dataset, we achieve competitive volume Dice scores of 0.963, 0.856 and 0.835 for kidney (including tumor and cysts), mass and tumor and surface Dice scores of 0.919, 0.749 and 0.729 for kidney (including tumor and cysts), mass and tumor respectively on our randomly-chosen validation cases.

Validation performance metrics indicate that the majority-vote segmentation is a reasonable proxy for individual reviewers' annotations. Both volumetric and surface Dice scores against individual annotations are quite similar to those attained on the majority predictions used for training. Nevertheless, alternative strategies leveraging the individual annotations directly, including ensemble prediction using networks trained by separate reviewers, may be of further interest. For applications where minimizing boundary error is particularly desirable, it may be beneficial to add a proxy for the surface Dice coefficient to the objective function directly.

Renal tumors and cysts remain challenging targets for segmentation due to their morphological heterogeneity and inconsistent Hounsfield intensity values between CT scans [7]. We have found that small masses are especially challenging for our current architecture, and that low tumor volume is associated with a decrease in both volumetric and surface Dice scores. Although U-Net archi-

tectures are known to perform robustly even with limited training data, it is possible that given a larger training set, higher-capacity models may achieve superior performance in renal mass segmentation. Given the relative abundance of publicly-available contrast-enhanced CT without voxel-level annotation, the design of semi-supervised or weakly-supervised architectures for 3D semantic segmentation is of particular interest to improve upon our current performance.

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
