# OpenReview forum: "Semantic Segmentation of Kidney, Cyst and Tumor"
_MICCAI.org/2021/Challenge/KiTS — Submitted to KiTS21 Challenge_

### Official Review · Reviewer_nsKM · 2021-08-30

**Rating:** 6

**Review:**

The authors present an approach that makes use of the nnU-Net after resampling to 2mm and sampling patches in such a way that oversamples "occupied" rather than background areas. It is not stated whether the authors made use of the multiple annotations per case or simply used the majority-voting approach. Beyond that, the paper has an adequate level of detail, but another figure would be nice, either to show predictions vs ground truth or a learning curve, etc.

---

### Official Review · Reviewer_ASey · 2021-08-30

**Rating:** 5

**Review:**

### Overall

- Once results are known, it would be nice to add a statement about how well it worked to the abstract

### Introduction

- There are some instances where spaces are omitted after sentences end, please fix those
- When referring to other works, please use the convention of "Lastname et al." rather than first name or last, first as you have done

### Methods

- Which method did you use to resample the imaging data? What about the segmentations?
- How did you chose your weighting in the loss function?

### Results

- It would be nice to show a figure with an example of your predictions here
- Please be sure to add the official results once they are known

### Discussion and Conclusion

- Please define the acronym "HU" before using it

---

### Decision · Program_Chairs · 2021-08-30

**Decision:**

Major Revisions

**Comment:**

Please address the reviewer comments and resubmit